# Motivators and barriers to mouthguard compliance by adult Gaelic football athletes

**Aoife Burke**[1,2]*, **Niall O'Connor**[2], **Niall Duffy**[2], **Katie Holohan**[2], **Enda F. Whyte**[1,2],
**Siobhán O'Connor**[1,2]

1 Centre for Injury Prevention and Performance, Athletic Therapy and Training, Dublin City University, Dublin, Ireland, 2 School of Health and Human Performance, Dublin City University, Dublin, Ireland

☯ These authors contributed equally to the work.
* aoife.burke@dcu.ie

## Abstract

### Introduction

Dental injuries contribute to 57% of reported maxillofacial injuries in Gaelic Football, with associated psychological and economic impacts on the affected athletes. Mouthguards have been developed in efforts to mitigate the incidence and severity of dental injuries, and use is mandatory in Gaelic Football. Dental claims have reduced by over 50% since mouthguards became mandatory, but costs of dental injuries are still prevalent. The aim of this study was to determine the mouthguard compliance rates in adult Gaelic Football players, as well as the motivations and barriers to compliance.

### Methods

This cross-sectional study utilised an online survey to determine the self-reported compliance of adult Gaelic Footballers with mouthguard use, the perceived peer compliance, and the motivations and barriers to compliance. Chi-square tests were used to examine differences between sex and between elite and sub-elite players.

### Results

A total of 545 Gaelic Footballers completed the survey. During training, 22% of players reported to always wear a mouthguard, with 48% never wearing it, and 30% occasionally wearing it. For games, 48% of players reported to wear a mouthguard, with 11% never wearing it and 41% occasionally wearing it. Motivating factors included teeth protection, gum protection and the rules of the game. The main barriers were discomfort, difficulty breathing and difficulty speaking. Females had significantly poorer compliance in training, but had significantly better compliance in games when compared to males.

### Conclusion

Mouthguard compliance is relatively poor amongst adult Gaelic Football players. Although compliance improves for games compared to training, there are still 1 in 2 players not wearing a mouthguard for games. Discomfort and challenges with breathing and speaking

**Data availability statement:** Relevant data are within the paper and its Supporting Information files. In addition, raw data is available in an OSF repository at the following link: (https://osf.io/kzwbt/).

**Funding:** The author(s) received no specific funding for this work.

**Competing interests:** The authors have declared that no competing interests exist.

suggest that players may benefit from having a custom-fit mouthguard. Coaches, refereeing officials and governing bodies should strive to implement the rules more often and improve education around the benefits of mouthguard use within the sport.

## Introduction

Gaelic Football is a high-intensity, high-velocity, multidirectional field sport that requires strength, speed and endurance [1]. It is estimated that 27% of Ireland's population participates in Gaelic Football [2], equating to roughly 472,500 adult players. Although Gaelic Football is an amateur sport, the physical demands exerted by the athletes are similar to those of professional soccer players [3], especially those who compete at an elite inter-county level [4]. Rules of the ladies game do not permit deliberate bodily contact between players, but males are permitted to use their bodies to confront opponents, when efforts are being made to dispossess the ball [5,6]. Due to the physical nature of the game, there is a high incidence of injury within the sport, with rates of 5.8 injuries per 1000 hours observed in males [7], and 7.9 injuries per 1000 hours in ladies Gaelic Footballers [1]. In 2021, maxillofacial injuries accounted for 5% of all adult male Gaelic Football injuries, with injuries to the teeth specifically contributing to 57% of these maxillofacial injuries [8]. While dental injury claims are lower in ladies Gaelic Football compared with males [8,9], exact figures on incidence rates in ladies Gaelic Football have not yet been published. Injuries to the teeth can be distressing with associated social, psychological and economic impacts on the affected athletes [10,11]. The management of these injuries can extend across an athlete's lifetime, with functional and aesthetic impairments associated with poor oral health-related quality of life outcomes in those affected by traumatic dental injuries [12].

Mouthguards have been introduced to sports in efforts to mitigate the incidence and severity of dental injuries [13]. Mouthguards increase the surface area over which the impact to the mouth is applied, acting to distribute the forces more widely and reducing the stress acting on one tooth alone [14]. A meta-analysis confirmed the protective effect of mouthguards, where the risk of traumatic dental injuries was 1.6 to 1.9 times greater when a mouthguard was not worn [15]. Not surprisingly, the Gaelic Games Association (GAA) introduced a rule in 2014, making mouthguard use mandatory for players of all grades in all male Gaelic Football games and training sessions. The Ladies Gaelic Football Association (LGFA) introduced the same rule into the ladies game in 2017. The introduction of this rule has observed a drop of 52% in the cost of dental injury claims in ladies Gaelic Footballers made via the LGFA injury fund, when compared to the three years prior to mandatory ruling of mouthguards in the game [16]. However, if dental claims are to be further reduced, it is important to determine the compliance rates of Gaelic Football participants in mouthguard use. With a mean cost of €968.54 per dental injury claim observed in a recent ladies Gaelic Football study [16], it is in the best interests of the GAA and LGFA to determine how compliance can be improved through the sport. A study looking at the compliance of children across a range of community sports, found 34% of participants wear a mouthguard voluntarily, but that compliance increased to 66% where a policy for mouthguard use existed [10]. This highlights the influence of protective equipment rules in sport. Nevertheless, the current compliance of mouthguard use in adult Gaelic Footballers has not yet been studied.

Across other team sports where mouthguard use is mandatory, compliance amongst adults has been found to be greatest in field hockey players (88–91%) [17,18], with lower rates observed in rugby players (54–68%) [19,20], and in ice hockey players (13–63%) [21,22]. Upon observation, compliance in elite sports has appeared greater within elite athletes (69%)

[19] compared to sub-elite athletes (54%) [20], but no studies to our knowledge have assessed differences between elite and sub-elite athletes. In order to improve compliance, it is pertinent to understand the motivations and barriers to mouthguard use. Although research has demonstrated a good understanding for mouthguard efficacy amongst athletes, this does not necessarily translate into sound compliance, with athlete behaviour often being influenced by peers [13]. Furthermore, the existing studies looking at mouthguard compliance in sport have often been self-reported, with little focus given to how peers perceive compliance amongst team-mates. This is important to consider as it may help to inform whether compliance may be affected by a greater peer culture within the game. Known barriers to mouthguard use across sports include discomfort [19], impedance with communication, challenges with breathing, and cost [23]. Therefore, the primary aim of this study is to determine the current compliance rates of, motivations and barriers to, mouthguard use in adult male and ladies Gaelic footballers. It is hypothesized that Gaelic Footballers will have poor compliance of mouthguard use in training, with greater compliance in games where rules are enforced. A secondary aim is to examine whether compliance rates, motivations and barriers differ between male and female players, and between elite and sub-elite players. It is hypothesized that females may demonstrate better compliance than males, and elite athletes will demonstrate better compliance than sub-elite athletes.

## Methods

### Study design and participants

This cross-sectional study was designed in line with the Strengthening the Reporting of Observational Studies in Epidemiology (STROBE) guidelines (Supplementary Material 1 in S1 File). Ethical approval for this project was granted by [redacted for blind review] research ethics committee, with written informed consent obtained from all participants prior to the commencement of the study. Current male and ladies adult Gaelic Footballers were recruited via emails sent to 160 local GAA club secretaries across the country (whose contact details could be sourced online), and via social media posts. Five clubs from each county (32 counties in total) were chosen at random to be contacted, so that geographic representation would be accounted for. Participants who no longer played Gaelic Football and those under the age of 18 were excluded from the study. Data was collected from the 29th of October 2021 to the 17th of December 2021. An a priori sample size calculation for a medium effect in a chi-square test was conducted using G*Power (Version 3.1). A minimum sample of 172 responses would be required to find a significant effect. A total of 568 athletes completed the survey. Responses were screened for completeness, and any responses with less than 100% of completion were removed from the analysis. Following removal of incomplete responses, the final sample consisted of 545 athletes. Elite Gaelic Footballers consisted of those competing at inter-county level, while sub-elite consisted of those competing at club or collegiate level [24].

An online survey (Google Forms, Google, California, USA) (Supplementary Material 2 in S1 File) was adapted to a Gaelic Football audience based on previously published literature [25]. The survey consisted of three sections. The first section captured details on the athlete demographics (age, sex, level of competition they are currently playing at, and the number of years playing Gaelic Football). The second section focused on self-reported mouthguard compliance, and perceived compliance of peers or teammates. The final section gathered information on the motivations and barriers for mouthguard compliance, and the perceived function of a mouthguard. A pilot of the survey was conducted with 6 research experts in the field of qualitative data for feedback, before being trialled on a group of recreational athletes (n = 10).

## Statistical analysis

Data analysis was performed in Excel Version 2202 (Microsoft Corporation Washington, USA) and SPSS Version 27 (IBM, Chicago, Illinois, USA). Frequency statistics were employed for assessing the compliance of mouthguard use in male and female Gaelic Footballers (during training and matches), and for exploring the motivations and barriers to mouthguard use.

Pearson Chi square (χ2) tests were used to examine the differences between male and female athletes, as well as examining the differences between elite and sub-elite athletes in mouthguard compliance, and the motivations and barriers to mouthguard use. A p-value of <0.05 indicated statistical significance and effect sizes were defined using the Phi (Φ) coefficient (small = 0.1, moderate = 0.3, large = 0.5) [26].

## Results

Five hundred and forty-five athletes completed the survey (367 males, 178 females). Participant demographics such as level of play, age and number of years playing Gaelic Football are presented in Table 1. No significant differences were observed for age or years playing between sexes.

With regards to self-reported mouthguard compliance, 26% of athletes reported to always wear a mouthguard during training, and compliance increased to 48% if athletes were playing a game. Over one third of athletes (36%) rarely or never wear a mouthguard during games, reporting to keep it in their sock until they are instructed to put it in by management staff or refereeing officials. A chi-square analysis revealed significant differences between male and female athlete compliance in training (χ2: 9.56, p: 0.048; Φ: 0.13 small effect size), whereby a greater proportion of females never wearing a mouthguard during training compared to males (Table 2; Fig 1). There was also a significant different between sexes in games (χ2: 15.13, p: 0.001; Φ: 0.17 small effect size), with a greater proportion of females always wearing

**Table 1. Group demographics and baseline differences between male and female athletes.**

| Competition | Level | Sex | % (n) | Age Mean ± SD (Years) | P Value^ | Years Playing Mean ± SD (Years) | P Value~ |
|---|---|---|---|---|---|---|---|
| Elite (n: 58) | Inter-County | Male | 60% (35) | 23.9 ± 4.6 | 0.06 | 17.9 ± 4.1 | 0.54 |
| | | Female | 40% (23) | 20.2 ± 1.9 | | 13.8 ± 3.3 | |
| | | Total | 100% (58) | 22.4 ± 4.2 | | 16.2 ± 4.3 | |
| Sub-Elite (n: 487) | Senior Club | Male | 46% (169) | 24.3 ± 5.7 | 0.14 | 17.2 ± 4.2 | 0.11 |
| | | Female | 30% (53) | 23.3 ± 3.6 | | 14.8 ± 4.9 | |
| | | Total | 100% (222) | 24.1 ± 5.3 | | 16.6 ± 4.5 | |
| | Intermediate Club | Male | 18% (66) | 24.7 ± 6.3 | 0.07 | 17.7 ± 3.9 | 0.12 |
| | | Female | 28% (50) | 23.3 ± 5.0 | | 15.0 ± 4.7 | |
| | | Total | 100% (116) | 24.1 ± 5.8 | | 16.5 ± 4.2 | |
| | Junior Club | Male | 22% (79) | 26.2 ± 7.1 | 0.97 | 17.4 ± 4.5 | 0.82 |
| | | Female | 22% (39) | 25.6 ± 7.0 | | 15.2 ± 4.9 | |
| | | Total | 100% (118) | 26.0 ± 7.0 | | 16.7 ± 4.7 | |
| | Collegiate | Male | 5% (18) | 21.2 ± 6.0 | 0.26 | 15.6 ± 3.2 | 0.32 |
| | | Female | 7% (13) | 19.7 ± 1.4 | | 15.2 ± 4.9 | |
| | | Total | 100% (31) | 20.6 ± 4.7 | | 14.3 ± 3.8 | |

N: Number of participants; SD: standard deviation; <: less than; ^: P Value indicating differences in age between sexes; ~: P Value indicating differences in number of years playing between sexes.

**Table 2. Self-reported compliance of mouthguard use for training and games in males and female, and in elite and sub-elite athletes.**

| | Always, 100% of the time | | More often than not, 75% of the time | | Sometimes, 50% - 75% of the time | | Rarely, I keep it in my sock | | Never wear it | | Chi-Square$^\Phi$ | P value |
|---|---|---|---|---|---|---|---|---|---|---|---|---|
| | Male % (n) | Female % (n) | Male % (n) | Female % (n) | Male % (n) | Female % (n) | Male % (n) | Female % (n) | Male % (n) | Female % (n) | | |
| **Training** | | | | | | | | | | | | |
| | 25% (90) | 16% (29) | 14% (53) | 11% (20) | 9% (31) | 8% (15) | 14% (51) | 12% (22) | 39% (142) | 52% (92) | 9.56 (0.13) | <0.05* |
| **Games** | | | | | | | | | | | | |
| | 44% (162) | 56% (100) | 9% (33) | 11% (19) | 5% (19) | 7% (13) | 28% (101) | 20% (36) | 14% (52) | 6% (10) | 15.13 (0.17) | 0.00* |
| | Elite % (n) | Sub-Elite % (n) | Elite % (n) | Sub-Elite % (n) | Elite % (n) | Sub-Elite % (n) | Elite % (n) | Sub-Elite % (n) | Elite % (n) | Sub-Elite % (n) | Chi-Square$^\Phi$ | P value |
| **Training** | | | | | | | | | | | | |
| | 21% (12) | 22% (107) | 26% (15) | 12% (58) | 9% (5) | 8% (41) | 7% (4) | 14% (69) | 38% (22) | 44% (212) | 9.99 (0.14) | 0.05 |
| **Games** | | | | | | | | | | | | |
| | 60% (35) | 47% (227) | 9% (5) | 10% (47) | 0% (0) | 7% (32) | 17% (10) | 26% (127) | 14% (8) | 11% (54) | 7.85 (0.12) | 0.10 |

N : Number of participants;

$^\Phi$: Phi coefficient effect size;

*: significant p value < 0.05.

a mouthguard in games compared to males (Table 2; Fig 1). No significant differences were found between elite and sub-elite athletes in self-reported mouthguard compliance (Table 2; Fig 2).

When asked to rate peer compliance, 73% of athletes perceived their peers to have very poor or poor compliance during training, and 7% perceived their peers to have good to excellent compliance. In games, 26% perceived their peers to have very poor or poor compliance, and 38% perceived peer compliance to be good or excellent.

Motivations and barriers to compliance are presented in Table 3. Popular motivators include "it's the rules of the game" (84% agreed), "injury insurance scheme" (80% agreed) and "dental care" (94% agreed). A chi-square analysis revealed "injury insurance" to be significantly more motivating to females compared to males ($\chi2$: 8.99, p: 0.01; $\Phi$: 0.17 small effect size) (Supplementary Material 3 in S1 File). The main barriers to compliance are "discomfort", "difficulty breathing" and "difficulty speaking". No significant differences were observed in motivations or barriers between elite and sub-elite athletes (Supplementary Material 4 in S1 File).

Lastly, when asked about their perceived function of a mouthguard, 100% of athletes agreed that it protected teeth, 66% agreed that it protected the jaw, 35% agreed that it may assist in reducing the effects of a concussion, and 67% agreed that it would protect the gums. No significant differences in perceived mouthguard function were observed between male and female athletes (Supplementary Material 3 in S1 File). A significantly greater proportion of elite athletes perceived the mouthguard to protect the jaw compared to sub-elite athletes ($\chi2$: 6.53, p: 0.04; $\Phi$: 0.11 small effect size) (Supplementary Material 4 in S1 File).

## Discussion

The primary aim of this study was to determine the compliance rate for mouthguard use amongst adult Gaelic Footballers. The primary hypothesis can be accepted as overall

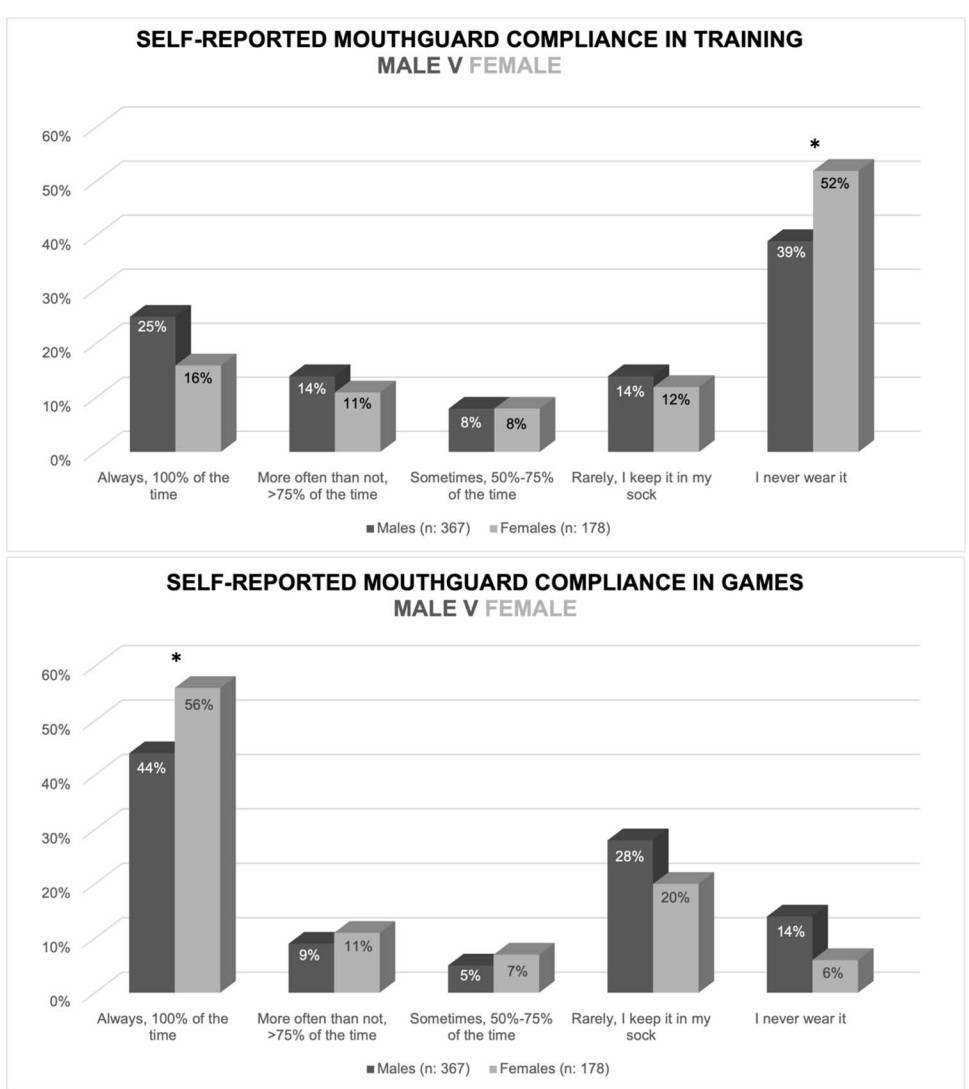

**Fig 1. Male and female self-reported compliance of mouthguard use (n = 545).** *: *Significant difference between males and females at p < 0.05.*

compliance rates were relatively poor, with the majority of athletes (56%) reporting to rarely or never wear their mouthguard during training. Self-reported compliance rates did increase for games as expected, with almost two-thirds of athletes reporting to wear it most of the time or always. These compliance rates align well with previous research [27], which found compliance rates of 54% and 43% in rugby [20] and field hockey [28] respectively. There are numerous reasons why compliance rates may be poorer in training than in games. Firstly, the intensity of training is typically lower than in games, with training sessions dedicating time to non-contact skills and low intensity player contact [29]. Consequently, injury rates are often lower in training than in games [30], and perhaps the majority of Gaelic Footballers perceive this lower intensity of play and lower injury risk as a reason for rarely or never wearing a mouthguard in training. Players appeared to recognise the poor compliance practices amongst their teammates, as observed in the findings of perceived peer compliance. This suggests a culture where not wearing a mouthguard is somewhat acceptable amongst peers, which may

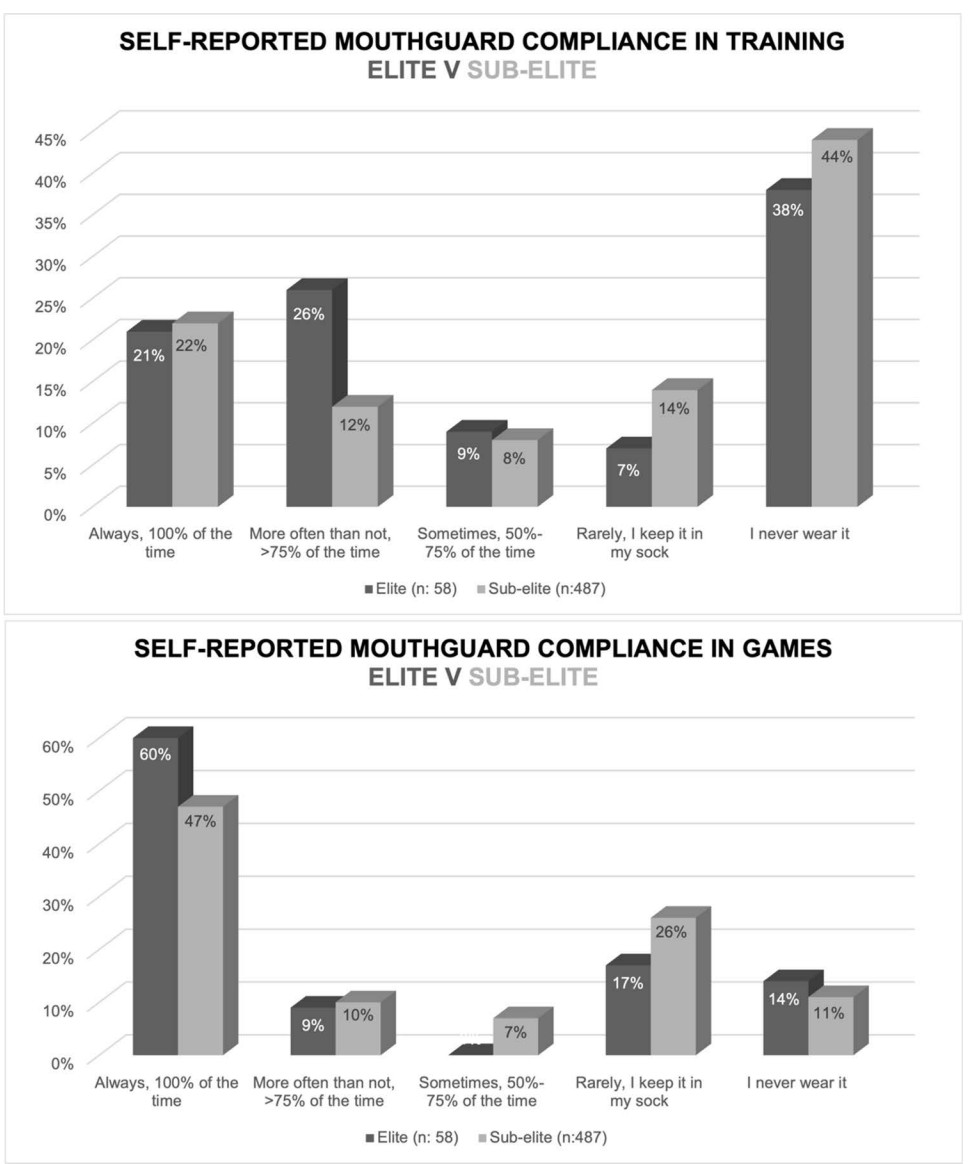

**Fig 2. Elite and sub-elite self-reported compliance of mouthguard use (n = 545).**

itself be an additional barrier to overcome. Secondly, there are no refereeing officials present at training sessions to enforce the mandatory mouthguard use rule, and therefore, Gaelic Footballers do not serve any punishment for their lack of compliance. Although management staff of sporting teams are aware of the rule and the benefits to mouthguard use, previous research has found that staff are not inclined to enforce mouthguard practices among athletes [21,31], highlighting the need for the GAA and LGFA to encourage coaches to ensure players wear mouthguards during all GAA and LGFA activities.

Another aspect relating to the primary aim of this study was to determine the motivations and barriers to mouthguard compliance in adult Gaelic Footballers. The majority of athletes with good compliance reported dental protection, rules of the game and injury insurance cover as the reasons for wearing a mouthguard most of the time or always. This is in line with

Table 3. Motivations, barriers and perceived function of mouthguards (n: 545).

|  | Agree | Neutral | Disagree |
|---|---|---|---|
|  | % (n) | % (n) | % (n) |
| **Motivations (n: 315)** |  |  |  |
| *Dental Care* | 94% (295) | 5% (16) | 1% (4) |
| *Rules of the Game* | 84% (263) | 12% (39) | 4% (13) |
| *Injury Insurance* | 80% (251) | 13% (40) | 8% (24) |
| *Reduce Concussion Effects* | 32% (100) | 42% (132) | 26% (83) |
| **Barriers (n: 230)** |  |  |  |
| *Difficulty Speaking* | 88% (202) | 8% (19) | 4% (9) |
| *Discomfort* | 80% (185) | 10% (24) | 9% (21) |
| *Difficulty Breathing* | 79% (182) | 14% (32) | 7% (16) |
| *Excess Saliva* | 46% (105) | 22% (50) | 33% (75) |
| *Dry Mouth* | 43% (99) | 24% (55) | 33% (76) |
| *Bad Taste/Odour* | 26% (59) | 28% (65) | 46% (106) |
| *Nausea* | 15% (35) | 20% (47) | 64% (148) |
| *Feel it is Unnecessary* | 12% (28) | 22% (51) | 66% (151) |
| *Aesthetics* | 10% (23) | 17% (40) | 73% (167) |
| *Cost* | 7% (15) | 20% (46) | 74% (169) |
| *Function (n: 545)* |  |  |  |
| *Teeth Protection* | 100% (543) | 0% (2) | 0% (0) |
| *Gum Protection* | 66% (362) | 23% (126) | 11% (57) |
| *Jaw Protection* | 66% (361) | 22% (120) | 12% (64) |
| *Reduce Concussion Effects* | 36% (194) | 39% (214) | 25% (137) |

N: Number of participants

previous research whereby player perception of efficacy in injury prevention largely dictates implementation practices [19]. The most common barriers to mouthguard use for those who rarely or never wear a mouthguard were difficulty speaking, discomfort and challenges with breathing. A recently published review on sports mouthguards has identified similar barriers to compliance [31]. Although the mouthguard fit (e.g., custom-made vacuum fit mouthguards) can influence comfort and perceived ease of breathing and communication [32], this alone does not seem enough to command good compliance [13]. A controlled randomised trial found that strength and performance were unaffected by custom-fit mouthguards, with athletes reporting the mouthguards to be comfortable and not causing difficulty breathing [33]. In contrast, strength and performance testing was negatively affected by the boil-and-bite mouthguard, with associated reports of discomfort and breathing challenges [33], highlighting the importance of mouthguard selection from athletes. Ultimately, education around mouthguard use appears to be the most advocated means of improving compliance [31,34], with lower compliance rates observed when education has not been provided [34]. The results of this study demonstrate a good understanding of mouthguard function amongst participants, especially with relation to dental, jaw and gum protection. Athletes seemed less sure of the potential benefits relating to concussion, and perhaps this is an avenue that should be explored more in player and coach education, especially if custom-fit mouthguards are implemented. Concussion injuries in sport not only result in time-loss from sport, but may also result in cognitive and emotional symptoms that can impair ability to work and study [35,36]. If custom-fit mouthguards have the potential to reduce the effects of concussion, in addition

to providing more comfort for the athlete, perhaps the GAA and LGFA could consider advocating or subsidising this type of mouthguard for its players.

A secondary aim of this study was to examine whether compliance rates, motivations and barriers differed between male and female players, and between elite and sub-elite players. The hypothesis that females may demonstrate better compliance than males can partly be accepted. Although a significantly larger proportion of females never wore a mouthguard in training, a significantly greater proportion of females always wore a mouthguard in games. Previous research has found females to be more compliant [37–39], but the majority of studies comparing sexes are based on teenagers or children [37,38]. A potential reason for greater compliance of females in games may be that females have been observed to have a higher sport morality than males [40]. In support of this, it is not surprising that "it's the rules of the game" and "dental care" were similarly ranked as top motivators for wearing a mouthguard in this study. With regards to elite vs sub-elite compliance, no significant differences were noted, and so the secondary hypothesis that speculated elite athletes to have better compliance than sub-elite athletes can be rejected.

## Practical recommendations

As demonstrated in this study, mouthguard compliance is relatively poor in both training and in games in adult Gaelic Football. Athletes are very much aware of the poor compliance, despite recognising the benefits to wearing a mouthguard. Based on the perceived barriers to compliance, discomfort and challenges with breathing and speaking appear to be the most prominent deterrents for use. Although it is difficult to balance the needs of the athlete without compromising the function of dental protection in the mouthguard, athletes should be encouraged to wear custom-fit mouthguards that are moulded by orthodontic specialists. Research has demonstrated these mouthguards to be less invasive on athlete comfort, breathing and communication. Perhaps custom-fit mouthguards should be subsidised so that they are more affordable and accessible to Gaelic Footballers. Secondly, further education could be provided to athletes on the costs of dental trauma and how quality of life can be affected after traumatic dental injuries. Within Gaelic Football, athletes are not covered by the Injury Insurance Scheme if they are not wearing a mouthguard at the time of injury. As a result, the athlete would be forced to fund the dental repair costs independently. With a mean cost of €968.54 per dental injury claim observed in a recent ladies Gaelic Football study [16], this is a significant sum to pay from an athlete's personal finances, and may act as an incentive to wear a mouthguard in the future. Coaches and management staff should check and promote mouthguard use amongst their teams. In addition, widespread education by the governing bodies (GAA and LGFA) is needed, to highlight the importance of mouthguard compliance and how a lack of compliance may prove costly if a dental injury occurs at a time when a mouthguard is not worn.

## Limitations

There are three main limitations to this study. Firstly, the survey did not capture details on the type of mouthguard that participants were using at the time. Custom-fit, standard-fit and boil-and-bite mouthguards have been noted to vary significantly in comfort and in functionality. Knowing the type of mouthguard used may have provided further insight into why compliance was relatively poor in this study. Secondly, the survey did not establish if athletes had sustained a traumatic dental injury in the past, which could have substantial implications on future engagement with mouthguard compliance. It has been reported that mouthguard compliance improves in those with a history of injury, but this was not captured in this study.

Thirdly, the main focus of this study was on adult Gaelic Football athletes, and did not capture the compliance of teenagers or children who make up a significant proportion of Gaelic Football athletes. Of note, mouthguard use only became compulsory in 2014 in the male game, and 2017 in the ladies game. At this time, the current participants of this study would have been teenagers and may not have been open to adapting their practices, especially if mouthguards were ill-fitting due to changes in maturation and growth occurring at a similar time. Due to ethical constraints, this study could only assess compliance in adults and thus future studies should assess compliance in adolescents and children. Perhaps compliance will improve as the years progress, especially where children who are currently participating will be encouraged to wear a mouthguard upon their first encounter with the sport. Lastly, this study employed self-reporting methods, which introduces a subjective bias.

Future studies should conduct spot-check observations of training sessions and games to more accurately determine compliance in both adult and minor populations, enhancing the accuracy of the results found in this study. Furthermore, details on the type of mouthguard owned by the athlete should be noted, alongside their history of dental injuries. These additional methodological enhancements would further develop our understanding of compliance and may potentially inform future policies and rules within the sport.

## Conclusion

Mouthguard compliance rates were relatively poor in training, with improved compliance observed in games. Males demonstrated greater compliance than females in training, but females had greater compliance in games. No differences in compliance rates were found between elite and non-elite athletes. Popular motivators for compliance included the rules of the game, dental protection and jaw protection. Discomfort, and challenges with breathing and communication were found to be the major barriers to compliance, suggesting a move towards custom-fit mouthguard advocation. Coaches, referees and governing bodies should strive to encourage compliance across all GAA and LGFA activities, in order to further reduce the incidence and cost of dental injuries within the sport.

## Supporting information

**S1 File. PLOS ONE Supplementary Material Mouthguard Compliance.**
(DOCX)

## Author contributions

**Conceptualization:** Aoife Burke.

**Data curation:** Niall O'Connor, Niall Duffy, Katie Holohan.

**Formal analysis:** Aoife Burke, Niall O'Connor, Niall Duffy, Katie Holohan.

**Methodology:** Aoife Burke.

**Supervision:** Aoife Burke.

**Writing – original draft:** Aoife Burke.

**Writing – review & editing:** Enda F Whyte, Siobhán O'Connor.

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
