## [Decision Letter · Decision Letter 0]

12 Feb 2025

PONE-D-24-54208Motivators and Barriers to Mouthguard Compliance by Adult Gaelic Football AthletesPLOS ONE

Dear Dr. Burke,

Thank you for submitting your manuscript to PLOS ONE. After careful consideration, we feel that it has merit but does not fully meet PLOS ONE’s publication criteria as it currently stands. Therefore, we invite you to submit a revised version of the manuscript that addresses the points raised during the review process.

We look forward to receiving your revised manuscript.

Kind regards,

Mario Lopes, Ph.D

Academic Editor

PLOS ONE

Journal Requirements:

Additional Editor Comments:

Congratulations to the Authors on their effort placed on their manuscript. The reviewers have made their decision on minor reviews.

Reviewers' comments:

Reviewer's Responses to Questions

**Comments to the Author**

1. Is the manuscript technically sound, and do the data support the conclusions?

Reviewer #1: Yes

Reviewer #2: Yes

2. Has the statistical analysis been performed appropriately and rigorously? 

Reviewer #1: Yes

Reviewer #2: Yes

3. Have the authors made all data underlying the findings in their manuscript fully available?

Reviewer #1: Yes

Reviewer #2: Yes

4. Is the manuscript presented in an intelligible fashion and written in standard English?

Reviewer #1: Yes

Reviewer #2: Yes

5. Review Comments to the Author

Reviewer #1: This manuscript investigates the motivators and barriers to mouthguard compliance among adult Gaelic football players, which is of significant practical importance. The study employs a cross-sectional survey design with a large sample size, effectively capturing the current status of mouthguard use among players. The use of chi-square tests to analyze differences between groups is appropriate. The results reveal relatively low compliance rates, gender disparities, and key motivators and barriers, providing a basis for future improvements.

However, there are several limitations in the study. Firstly, the survey did not collect information on the specific types of mouthguards used by the players. Different types of mouthguards (e.g., standard-fit, boil-and-bite, custom-fit) vary significantly in comfort and functionality, which could be a crucial factor influencing compliance. Secondly, the study did not inquire about players' history of dental injuries, which may substantially affect their motivation to wear mouthguards. Thirdly, the study focused solely on adult players and did not include adolescents or children, who also constitute a significant portion of Gaelic football participants and may have different compliance patterns. Lastly, relying solely on self-reported data from players may introduce subjective biases. It is recommended to incorporate observational methods to enhance the accuracy of the results.

Despite these limitations, the study provides valuable insights into mouthguard use in Gaelic football. The authors are encouraged to address the aforementioned limitations in future research by including data on mouthguard types, injury history, and compliance among younger players. Additionally, employing multiple research methods could reduce subjective biases. The study is worth publishing but would benefit from further refinement in subsequent research to enhance the comprehensiveness and reliability of the findings.

Reviewer #2: This is an interesting and well-written study. To give more context to the study, it would have been interesting to mention how many people practice Gaelic football.

It would also add, if possible, a comparison with similar studies done for other sports are reported and if specific behavior could be identified for Gaelic football practitioners compared to other sports concerning mouthguard-wearing habits.

6. PLOS authors have the option to publish the peer review history of their article (what does this mean? ). If published, this will include your full peer review and any attached files.

**Do you want your identity to be public for this peer review?** For information about this choice, including consent withdrawal, please see our Privacy Policy .

Reviewer #1: No

Reviewer #2: No

---

## [Author Response · Author response to Decision Letter 0]

24 Feb 2025

Dear Editor and Reviewers,

Firstly, the authors would like to thank you sincerely for taking the time and effort to provide us with your insight in reviewing our manuscript. We understand that this is completely voluntary, and would really like to express our gratitude. The authors have made some suggested changes within the manuscript, and have outlined those changes in addition to responses under Table 1 in the "Response to Reviewers" document.

Please do not hesitate to contact me, should there be any queries or mis-understandings.

Many thanks in advance.

Best wishes,

Aoife Burke

---

## [Editor Report · Decision Letter 1]

2 Mar 2025

Motivators and Barriers to Mouthguard Compliance by Adult Gaelic Football Athletes

PONE-D-24-54208R1

Dear Dr. Aoife Burke,

We’re pleased to inform you that your manuscript has been judged scientifically suitable for publication and will be formally accepted for publication once it meets all outstanding technical requirements.

Kind regards,

Mario Lopes, Ph.D

Academic Editor

PLOS ONE
---

## [Editor Report · Acceptance letter]

PONE-D-24-54208R1

PLOS ONE

Dear Dr. Burke,

I'm pleased to inform you that your manuscript has been deemed suitable for publication in PLOS ONE. Congratulations! Your manuscript is now being handed over to our production team.

Kind regards,

on behalf of

Prof. Mario Lopes

Academic Editor

PLOS ONE